# Scalable and Efficient Non-adaptive Deterministic Group Testing

**Dariusz R. Kowalski**
School of Computer and Cyber Sciences
Augusta University, USA
dkowalski@augusta.edu

**Dominik Pajak**
Department of Pure Mathematics
Wroclaw University of Science and Technology,
Infermedica, Poland
dominik.pajak@pwr.edu.pl

## Abstract

Group Testing (GT) is about learning a (hidden) subset $K$, of size $k$, of some large domain $N$, of size $n \gg k$, using a sequence of queries. A result of a query provides some information about the intersection of the query with the unknown set $K$. The goal is to design efficient (polynomial time) and scalable (polylogarithmic number of queries per element in $K$) algorithms for constructing queries that allow to decode *every* hidden set $K$ based on the results of the queries. A vast majority of the previous work focused on randomized algorithms minimizing the number of queries; however, in case of large domains $N$, randomization may result in a significant deviation from the expected precision of learning the set $K$. Others assumed unlimited computational power (existential results) or adaptiveness of queries (next query could be constructed taking into account the results of the previous queries) – the former approach is less practical due to non-efficiency, and the latter has several drawbacks including non-parallelization. To avoid all the abovementioned drawbacks, for Quantitative Group Testing (QGT) where query result is the size of its intersection with the hidden set, we present *the first* efficient and scalable *non-adaptive deterministic* algorithms for constructing queries and decoding a hidden set $K$ from the results of the queries – these solutions *do not* use any randomization, adaptiveness or unlimited computational power.

## 1 Introduction

In the Group Testing (GT) field, the goal is to identify all elements of an unknown set $K$ by asking queries. Originally GT was applied to identifying infected individuals in large populations using pooled tests [26] which regained interest during COVID-19 pandemic [2, 52, 56]. Recently GT found applications in various areas of Machine Learning including approximating the nearest neighbor [28], simplifying multi-label classifiers [58] and accelerating forward pass of a deep neural network [51].

All we initially know about set $K$ is that $|K| \leq k$, for some known parameter $k \leq n$, and that it is a subset of some much larger set $N$ with $|N| = n$. The answer to a query $Q$ depends on the intersection between $K$ and $Q$ and equals to $\mathsf{Result}(K \cap Q)$, where $\mathsf{Result}$ is some result function (also called feedback function in this paper).

The sequence of queries is a correct solution to Group Testing if and only if for any two different sets $K_1, K_2$ (satisfying some cardinality restriction), the sequence of answers for $K_1$ and $K_2$ is different. Note that although this allows to uniquely identify the hidden set $K$ based on the results of the queries, in some cases decoding of set $K$ could be a hard computational problem. The objective is: for a given deterministic feedback function $\mathsf{Result}(\cdot)$, to find a fixed sequence of queries that will identify any set $K$, and the length of this sequence, called the query complexity, is as short as possible. In particular, we are interested in solutions that have query complexity logarithmic in $n$ and polynomial in $k$.

36th Conference on Neural Information Processing Systems (NeurIPS 2022).

| $\alpha$ | Constructive algorithms | Lower bound |
|---|---|---|
| 1 | $O(k^2 \log n)$ [54] | $\Omega\left(k^2 \frac{\log n}{\log k}\right)$ [12] |
| $k$ | $O(k^2 \log n)$ [54] 
 $\widetilde{O}(k)$ **Thm 1** (for $\alpha = k$) | $\Omega\left(k \frac{\log n}{\log k}\right)$ (folklore) |
| $*$ | $\widetilde{O}\left(\min\left\{\left(\frac{k}{\alpha}\right)^2, \frac{n}{\alpha}\right\} + k\right)$ (**Thm 1**) | $\Omega\left(\min\left\{\left(\frac{k}{\alpha}\right)^2, \frac{n}{\alpha}\right\} + k\frac{\log n}{\log k}\right)$ (**Thm 2**) |

Table 1: Asymptotic bounds on query complexity of solutions to non-adaptive deterministic QGT with $\mathcal{F}_\alpha$ feedback. By constructive upper bound we mean query construction and decoding of hidden set in time $poly(n)$. Symbol $*$ stands for any valid value of the parameter $\alpha \in \{1, \ldots, k\}$, notations $\widetilde{O}$ and $\widetilde{\Omega}$ disregard polylogarithmic factors.

The most popular classical variant, present in the literature, considers function Result$(\cdot)$ that answers whether the intersection between $K$ and $Q$ is empty or not, c.f., [27]; it is also known under the name of *beeping*. Another popular result function returns the intersection size; this variant has also been studied under the name of *coin weighting* [4, 24] and Quantitative Group Testing (QGT) [32, 29].

In this paper we study the problem of Group Testing under a more general *capped quantitative result*, where the result (feedback) is the size of the intersection up to some parameter $\alpha \in \{1, \ldots, k\}$, and $\alpha$ for larger intersections. It subsumes and generalizes the two previously described classical result functions: the smallest case of $\alpha = 1$ corresponds to the classical empty/non-empty feedback (beeping), while the case $\alpha = k$ captures the (full) quantitative feedback (classical QGT). Motivation for our research is twofold: first, to better understand the intermediate settings between the two classical GT models; second, to find efficient solutions for the capped settings that could be applied to other problems (see the Appendix).

Our focus is on *non-adaptive deterministic* solutions, in which queries are fixed (i.e., the same sequence applies to all hidden sets) and allow to discover (decode) any hidden set based on $\alpha$-capped results. The primary goal is to minimize the number of queries, called *query complexity*, and the second objective is to compute the queries and decode the hidden set in polynomial time. All existing polynomial construction/decoding algorithms for beeping or quantitative feedbacks use $\Theta(\min\{n, k^2 \log n\})$ queries, c.f., [54]. General constructions developed in this work employ the concept of *dispersion*, c.f., [57], and combine it appropriately with other techniques. When instantiated for a specific feedback parameter $\alpha \in \{1, \ldots, k\}$, (e.g., $\alpha = \sqrt{k}$), they shrink the gap between the upper and lower bounds on the number of efficiently constructed queries nearly exponentially: from linear to polylogarithmic gap per hidden element. Together with our lower bound, they explain why sometimes a much smaller feedback is sufficient for decoding sets with similar efficiency. We show that parameter $\alpha$ has also different interpretations – it inversely relates to the number of occurrences of an element in the queries and to fault-tolerance (c.f., Section 6).

## 1.1 Our Results (see also Table 1)

We generalize the classical beeping and QGT models of Group Testing by considering an $\alpha$-*capped results*, also called an $\alpha$-*capped feedback* function, for an $\alpha \in \{1, \ldots, k\}$: for a hidden set $K$ and query $Q$ it returns $\mathcal{F}_\alpha(Q \cap K) = \min\{|Q \cap K|, \alpha\}$. We study the query complexity of non-adaptive deterministic QGT in this general model. We focus on scalable (polylogarithmic number of queries per element) and efficient (polynomial-time) constructing/decoding algorithms, and we show that it could be achieved *if and only* if the information cap $\alpha = \widetilde{\Omega}(\sqrt{k})$. In particular, it holds for classical QGT.

**Main result – Polynomial-time construction/decoding algorithms using almost optimal number of queries (Algorithms 1 and 2 in Section 4).** Here almost-optimality means that the length of the constructed query sequence is only polylogarithmically longer than the shortest possible sequence. The previous best polynomial-time solution used $\Theta(\min\{n, k^2 \log n\})$ queries for all $\alpha \geq 1$ [54], and we shrink it by factor $\Theta(\min\{\alpha^2, k\} \text{polylog}^{-1} n)$. To achieve this goal, we define and build new types of selectors, called (Strong) Selectors under Interference (SSuI and SuI, for short) based on the concept of (superimposed) dispersion. Although dispersion has been used before in GT, c.f., [10, 44], as well as the classical superimposed codes, our technical novelty is twofold: first, we identified additional properties of selectors (e.g., avoiding interfering set with capped intersections), that could

not be formally deducted from the definitions of previous types of selectors, but allow to improve the solutions to QGT; second, we applied careful mathematical analysis to choose specific parameters of dispersers and superimposed codes and to prove that the new selectors' properties hold.

We also generalize the concept of Round-Robin query systems, where each query is a singleton, to $\alpha$-Round-Robin query systems, containing sets of size at most $\alpha$. Such sequences are shorter than the simple Round-Robin, *i.e.,* have length $O((n/\alpha)\,\text{polylog}\,n)$, and, unlike a simple Round-Robin singletons' structure, are challenging to construct in a way to allow correct decoding based on $\alpha$-capped feedback.

**Theorem 1.** *There is an explicit polynomial-time algorithm constructing non-adaptive queries* $Q_1, \ldots, Q_m$, *for* $m = O\left(\min\left\{\left(\frac{k}{\alpha}\right)^2, \frac{n}{\alpha}\right\}\,\text{polylog}\,n + k\,\text{polylog}\,n\right)$, *that solve Group Testing under feedback* $\mathcal{F}_\alpha$ *with polynomial-time decoding. Moreover, every element occurs in* $O(\frac{k}{\alpha}\,\text{polylog}\,n)$ *queries, and the decoding time is* $O(m + \frac{k^2}{\alpha}\,\text{polylog}\,n + k\,\text{polylog}\,n)$.

One of the consequences is that having the result function capped at $\alpha = \sqrt{k}$, we obtain similar number of queries as with the actual result (i.e., returning the size of the whole intersection, up to $k$), which together with our next lower bound establishes an interesting inherited property of GT. The value $\alpha = \sqrt{k}$ seems to be a sweet spot balancing two different phenomena – superimposed coding and dispersion. Enriching the beeping model (i.e., $\alpha$ increases starting from 1) allows to improve the superquadratic query complexity of classical superimposed coding (used for the beeping feedback), and the improvement is roughly quadratic in terms of $\alpha$, due to the nature of superimposed codes. On the other hand, limiting the full quantitative model (i.e., $\alpha$ decreases starting from $k$) increases the number of queries, starting from slightly superlinear, roughly linearly in $\alpha$, as the nature of dispersion is linear. Both these phenomena equalize query complexities at $\alpha = \sqrt{k}$. This is illustrated in our construction – in the beginning, the progress in learning relies on dispersion-based SuI's for small $\alpha$, then with the growth of $\alpha$ the length of used SuIs becomes similar to the length of the code-based SSuI; it happens for $\alpha = \sqrt{k}$. The SSuI allows to discover all the remaining elements and finishes the learning process. The last fact explains why further increase of $\alpha$ is not needed.

**Lower bound (Section 5).** The almost-optimality of our algorithms from Theorem 1 is justified by proving an absolute lower bound on the length of sequences allowing to decode a hidden set based on feedback $\mathcal{F}_\alpha$ to the queries. Here by "absolute" we mean that it holds for all query systems that allow for decoding of the hidden sets based on feedback $\mathcal{F}_\alpha$, not restricted to polynomially constructed queries with polynomial decoding algorithm. Even more, some components of the lower bound are general: they hold for *any $\alpha$-capped* feedback function, which will be formally defined later in Section 3. The lower bound has three components: $(k/\alpha)^2$, $n/\alpha$, and $k\frac{\log \frac{n}{k}}{\log \alpha}$. For different ranges of $k$, different components determine the value of the lower bound. Note that these components match the corresponding components in our constructive upper bound (Main result in Theorem 1), up to a polylogarithmic factor.

**Theorem 2.** *Any non-adaptive algorithm solving Group Testing needs:*

- $\Omega\left(\min\left\{\left(\frac{k}{\alpha}\right)^2, \frac{n}{\alpha}\right\} + k\frac{\log \frac{n}{k}}{\log \alpha}\right)$ *queries under feedback* $\mathcal{F}_\alpha$.

- $\Omega\left(\min\left\{\left(\frac{k}{\alpha}\right)^2, \frac{n}{\alpha}\right\}\right)$ *queries under any feedback capped at* $\alpha$.

**Document structure.** We discuss related work on various variants of GT in Section 2. In Section 3 we formally define the QGT problem and the generalized $\alpha$-capped-results model. In Section 4 we present our polynomial-time query construction and decoding algorithms. Section 5 proves the lower bound. Discussion of extensions, applications and future directions is given in Section 6. Additional Appendix contains more details: Appendix A discusses other related work; Appendix B shows details of extensions and applications; Our new selector tools are constructed and analyzed in Appendix C.

## 2 Previous and Related Work (see also Tables 1 and 2)

Group Testing with general feedback functions have been studied in [47], where two parameters of feedback functions (cap and expressiveness) are shown to influence the necessary number of queries.

In the standard feedback model, considered in most of the Group Testing literature [27], the feedback tells whether the intersection between query $Q$ and set $K$ is empty or not (sometimes it is also

| $\alpha$ | Upper bound | First work with asymptotic bound | Difference from this paper |
|---|---|---|---|
| $k$ | $O\left(k\frac{\log n}{\log k}\right)$ | [37] | existential (super-exponential construction time) |
| | $O\left(k\frac{\log n}{\log k}\right)$ | [4] | adaptive |
| | $O\left(k\frac{\log n}{\log k}\right)$ | [55] | randomized (only with some probability) |

Table 2: Other related work on *adaptive*, *randomized* or *existential* solutions to QGT for $\alpha = k$.

called a *beeping model*). It it is a special case $\mathcal{F}_1$ of our feedback function. In this feedback model, Group Testing is known to be solvable using $O(k^2 \log(n/k))$ queries [21] and an explicit polynomial-time construction of length $O(k^2 \log n)$ exists [54]. Best known lower bound (for $k < \sqrt{n}$) is $\Omega(k^2 \log n / \log k)$ [12]. This model is also similar to Angluin's notion of concept learning with disjointness queries [1]. The main differences are that in our model the learner is non-adaptive and that in the disjointness queries the feedback might include an element from the intersection.

The setting considered in this paper is also a generalization of an existing problem of coin weighting. In this problem, we have a set of $n$ coins of two distinct weights $w_0$ (true coin) and $w_1$ (counterfeit coin), out of which up to $k$ are counterfeit ones. We are allowed to weigh any subset of coins on a spring scale, hence we can deduce the number of counterfeit coins in each weighting. The task is to identify all the counterfeit coins. Such a feedback is a special case $\mathcal{F}_k$ of our feedback function. The problem is solvable with $O(k \log(n/k)/ \log k)$ [37] non-adaptive (i.e., fixed in advance) queries and matching a standard information-theoretic lower bound of $\Omega(k \log(n/k)/ \log k)$, as well as its stronger version proved for randomized strategies [24]. [4] considers the problem of explicit polynomial-time construction of $O(k \log(n/k)/ \log k)$ queries that allows for polynomial time identification of the counterfeit coins. However, the algorithm presented in [4] is adaptive, which means that the subsequent queries can depend on the feedback from the previous ones. The only existing, constructive, non-adaptive solution would be using the explicit construction of the superimposed codes [46] but the resulting query complexity would be $O(k^2 \text{polylog } n)$. Thus, the solution in our paper is the first explicit polynomial time algorithm constructing non-adaptive queries allowing for fast decoding of set $K$, with $O(k \text{ polylog } n)$ fixed queries.

An important line of work on non-adaptive *randomized* solutions to Quantitative Group Testing, started by [55] and being continued until recently [32, 14, 29, 3], resulted in a number of algorithms nearing the lower bound of $2k\frac{\ln(n/k)}{\ln k}$ from [25]. However, these results always assume some restriction on $k$ (typically $k \sim n^\theta$ for some $0 < \theta < 1$), and similarly as above, they may result in significant deviation from the actual set if $n$ is large. They also require a large number of truly random bits (in practice, algorithms have access only to small truly random seeds, while randomized solutions typically have large number of $\Omega(n \log n \log k)$ random bits to construct the queries) and typically do not offer any provable guarantees in case when an adaptive adversary could create/change the hidden set or introduce errors online during the learning process.

The bounds obtained in this paper match (up to polylogarithmic factors) the best existing results for the extreme cases of $\alpha = 1$ and $\alpha = k$. The paper also bounds how the query complexity depends on the value of $\alpha$ between these extremes. Interestingly we show, that the shortest-possible query complexity of $k \text{ polylog } n$ is already possible for $\alpha = \sqrt{k}$ and increasing $\alpha$ from $\sqrt{k}$ to $k$ does not result in further decrease of the query complexity.

Other interesting feedback function is a *Threshold Group Testing* [20, 23], where the feedback model includes a set of thresholds and the feedback function returns whether or not the size of the intersection is larger or smaller than each threshold. In [22] the authors show that it is possible to define an interval of $\sqrt{k \log k}$ thresholds resulting in an algorithm with $O(k \log(n/k)/ \log k)$ queries. Note that both those feedbacks are "inefficient" in view of our setting of $\alpha$-capped feedbacks, because their feedback functions are not capped at any $\alpha < k$, but they achieve similar query complexity as our capped $\mathcal{F}_{\sqrt{k}}$ feedback.

Superimposed codes and dispersers were already used in Group Testing. However, they either led to a super-quadratic (in $k$) number of queries, c.f., [46, 9], or decoded only a fraction of elements of the hidden set, see [44, 41]. Recall that in solutions where query complexity depends on the number of identified elements, decoding of all the elements requires over $k^2$ queries, as proved in [21, 10]. [45]

presents the first Group Testing solution with $poly(k, \log n)$ decoding time, but super-quadratic query complexity. It is worth noting that our generalized solution achieves almost-linear number of queries (for certain values of $\alpha$) and our decoding algorithm identifies all the elements in time polynomial in $k$ and logarithmic in $n$. Our use of the known tools is different then in previous approaches: we define new properties (SuI, SSuI), which we prove to be satisfied by some combinations of those tools, and lead to efficient solutions in both query complexity and construction/decoding time.

## 3 The Model and the Problem

We assume that the universe of all elements $N$, with $|N| = n$, is enumerated with integers $1, 2, \ldots, n$. Throughout the paper, we will associate an element with its identifier. Let $K$, with $|K| \leq k$, denote a hidden subset of $N$ chosen arbitrarily by an adversary. Let $\mathcal{Q} = \langle Q_1, \ldots, Q_m \rangle$ be a non-adaptive algorithm, represented by a sequence of $m$ queries fixed prior to an execution.

A general feedback function $\mathcal{F}$ is a function from subsets of $N$ into an arbitrary domain. A function is applied to $K \cap Q$. A meaningful feedback function can range from simple empty/non-empty feedback up to returning all elements of the intersection. Consider a feedback function $\mathcal{F}_\alpha$, that returns the size of an intersection if it is at most $\alpha$ and $\alpha$ for larger intersections, *i.e.,* $\mathcal{F}_\alpha(Q \cap K) = \min\{|Q \cap K|, \alpha\}$. Parameter $\alpha$ in feedback $\mathcal{F}_\alpha$ is called a *feedback cap*. A general class of feedback functions (used in our lower bound) with feedback cap $\alpha$ includes all deterministic functions that take subsets of $N$ as input and for sets with more than $\alpha$ elements output some arbitrary fixed value.

We say that $\mathcal{Q}$ solves Quantitative Group Testing (QGT), if the feedback vector allows for unique identification of set $K$. The feedback vector is defined as $\langle \mathcal{F}_\alpha(Q_1 \cap K), \mathcal{F}_\alpha(Q_2 \cap K), \ldots, \mathcal{F}_\alpha(Q_t \cap K) \rangle$. Thus, in order to solve QGT, the feedback vectors for any two sets $K_1$ and $K_2$ have to be different. We say that $\mathcal{Q}$ solves QGT with polynomial-time reconstruction if there exists a polynomial-time algorithm that, given the feedback vector outputs all the identifiers of the elements from $K$. Finally we say that $\mathcal{Q}$ is constructible in polynomial time if there exists a polynomial-time algorithm, that given parameters $n, k, \alpha$ outputs an appropriate sequence of queries.

We assume that both coupled algorithms, construction and decoding, know $n, k, \mathcal{F}_\alpha$. W.l.o.g., in order to avoid rounding in the presentation, we assume that $n$ and other crucial parameters are powers of 2.

## 4 Polynomial-time Constructions and Decoding

### 4.1 Combinatorial Tools

In this section we introduce two new combinatorial tools (Selectors-under-Interference and Strong-Selectors-under-Interference) for our QGT solutions, and recall Dispersers and Balanced IDs.

For given sets $K_1, K_2 \subseteq N$ and an element $v \in K_1$, we say that $S \subseteq N$ *selects $v$ from $K_1$ under $\alpha$-interference from $K_2$* if $S \cap K_1 = \{v\}$ and $|S \cap K_2| < \alpha$. Intuitively, $v$ is a unique representative of $K_1$ in $S$ and the number of representatives of $K_2$ in $S$ is smaller than $\alpha$.

**Definition 1** (Selector under Interference (SuI))**.** *An $(n, \ell, \epsilon, \kappa, \alpha)$-Selector-under-Interference, $(n, \ell, \epsilon, \kappa, \alpha)$-SuI for short, is a sequence of queries $\mathcal{S} = (S_1, \ldots, S_x)$ satisfying: for every set $K_1 \subseteq N$ of at most $\ell$ elements and set $K_2 \subseteq N$ of at most $\kappa$ elements, there are at most $\epsilon \ell$ elements $v \in K_1$ that are not selected from $K_1$ under $\alpha$-interference from $K_2$ by any query $S_i \in \mathcal{S}$, i.e., set $\{v \in K_1 : \forall_{i \leq x} S_i \cap K_1 \neq \{v\} \text{ or } |S_i \cap K_2| \geq \alpha\}$ has less than $\epsilon \ell$ elements.*

**Disperser.** Consider a bipartite graph $G = (V, W, E)$, where $|V| = n$, which is an $(\ell^* = \epsilon \ell, d, \epsilon)$-*disperser with entropy loss $\delta$*, i.e., it has left-degree $d$, $|W| = \ell d/\delta$, and satisfies the following dispersion condition: for each $L \subseteq V$ such that $|L| \geq \ell^*$, the set $N_G(L)$ of neighbors of $L$ in graph $G$ is of size at least $(1 - \epsilon)|W|$. Note that it is enough for us to take as $\epsilon$ in the dispersion property the same value as in the constructed $(n, \ell, \epsilon, \kappa, \alpha)$-SuI $\mathcal{S}$.[1] An explicit construction (i.e., in time polynomial in $n$) of dispersers was given by [57], for any $n \geq \ell$, and some $\delta = O(\log^3 n)$, where $d = O(\text{polylog } n)$.[2] In the following the notation $d$ and $\delta$ will always denote the parameters of the disperser. Note that any improved construction of dispersers will immediately reduce the complexities of our algorithms. In Appendix C.1 we will describe two disperser-based polynomial-time constructions of SuI and prove:

---

[1]If someone considers $\epsilon \geq 1/3$, we could use construction for $\epsilon = 1/3$ to get solution with better guarantees.

[2]Optimal dispersers have $\delta = O(1)$ and $d = O(\log n)$, but their polynomial-time construction is an open problem.

**Theorem 3.** *There is an explicit polynomial-time construction of an $(n, \ell, \epsilon, \kappa, \alpha)$-SuI, for any $\ell$, any $\alpha \leq k$ such that $\ell\alpha/(3\delta) > \kappa$, for any constant $\epsilon \in (0, 1/3)$, of size $O(\min\{n, \ell d\delta \log^2 n\})$. Moreover, every element occurs in $O(d\delta \log n)$ queries.*

**Theorem 4.** *There is an explicit polynomial-time construction of an $(n, \ell, \epsilon, \kappa, \alpha)$-SuI, for any $\ell$, any $\alpha \leq k$ such that $\ell \leq 3\delta\kappa/\alpha$ for any constant $\epsilon \in (0, 1/3)$, of size $O\left(\min\{n, \frac{\kappa d\delta}{\alpha}\log^2 n\} + \frac{n d\delta}{\alpha}\log n\right)$. Moreover, every element occurs in $O(d\delta \log n)$ queries.*

**Definition 2** (Strong Selector under Interference (SSuI)). *An $(n, \ell, \kappa, \alpha)$-Strong-Selector-under-Interference, $(n, \ell, \kappa, \alpha)$-SSuI for short, is a sequence of queries $\mathcal{T} = (T_1, \ldots, T_x)$ satisfying: for every set $K_1 \subseteq N$ of at most $\ell$ elements and set $K_2 \subseteq N$ of at most $\kappa$ elements, every element $v \in K_1$ is selected from $K_1$ under $\alpha$-interference from $K_2$ by some query $T_i \in \mathcal{T}$, i.e., set $\{v \in K_1 : \forall_{i \leq x} T_i \cap K_1 \neq \{v\} \text{ or } |T_i \cap K_2| \geq \alpha\}$ is empty.*

An $(n, \ell, \kappa, \alpha)$-Strong-Selector-under-Interference could be also viewed as $(n, \ell, 0, \kappa, \alpha)$-Selector-under-Interference. In Appendix C.2 we describe a polynomial-time construction of SSuI, which essentially is a [46] construction for adjusted parameters, and prove that it satisfies the SSuI property.

**Theorem 5.** *There is an explicit polynomial-time construction of an $(n, \ell, \kappa, \alpha)$-SSuI of length $O(\ell^2 \log_\ell^2 n)$, provided $\ell \geq 3\delta\kappa/\alpha$. Moreover, every element occurs in $O(\ell \log_\ell n)$ queries.*

**Balanced IDs.** Each element $i$ in $N$ has a unique ID represented by $2\log_2 n$ bits, in which the number of 1's is the same as the number of 0's; e.g., take a binary representations of elements $i$ and $n - i$, each in $\log_2 n$ bits, and concatenate them (recall $n = |N|$). Balanced IDs have previously been used in algorithms for decoding elements in Group Testing (see e.g., [50]).

### 4.2 Construction of Queries, Decoding and Analysis (Proof of Theorem 1)

**Algorithm constructing queries.** Assume $\alpha \geq 2$ and $k > 3\delta k/(\alpha - 1)$ (the opposite case will be considered later) and let us take $(n, \ell, 1/2, k, \alpha - 1)$-SuI $\mathcal{S}^{(\ell)}$, for $\ell$ being a power of 2 ranging down from $k$ to $3\delta k/(\alpha - 1)$ (w.l.o.g. we could also assume that $3\delta k/(\alpha - 1)$ is a power of 2). Next, for each set $S$ in these selectors we add the following family $\mathcal{R}(S) = \{R_i(S)\}_{i=1}^{2\log_2 n}$ of sets $R_i(S) = \{v \in S : \lfloor v/2^{i-1} \rfloor = 1 \mod 2\}$. Intuitively $R_i(S)$ is the set of elements from $S$ that have 1 on $i$-th least significant bit of Balanced ID. Let us call the obtained enhanced selectors (i.e., with additional families $\mathcal{R}(S)$, for every set $S$ in the original selector) $\bar{\mathcal{S}}^{(\ell)}$. Then we concatenate selectors $\mathcal{R}(\mathcal{S}^{(\ell)})$, starting from the largest $\ell = k$, to the smallest value $\ell = 3\delta k/(\alpha - 1)$. An $(n, 3\delta k/(\alpha - 1), k, \alpha - 1)$-SSuI $\mathcal{T}$ is concatenated at the end, with the same replacement of bits 1 and 0 that gives $\mathcal{R}(\mathcal{T})$ as in the above $(n, \ell, 1/2, k, \alpha - 1)$-SuI's. In the case, where $\alpha = 1$ or $k \leq 3\delta k/(\alpha - 1)$ then the sequence equals to $(n, k, k, \alpha)$-SSuI $\mathcal{T}$ concatenated as in the above with $\mathcal{R}(\mathcal{T})$. Algorithm 1 presents a pseudocode of the construction algorithm for $\alpha > 1$.

---

**Algorithm 1:** Construction of a sequence of queries solving QGT with $\alpha$-capped feedback.

**Input:** Disperser $G$ of degree $d$ and entropy loss $\delta$

```
1  ℓ ← k, Q ← ⟨⟩;
2  while ℓ > 3δk/(α−1) do
3      S ← (n, ℓ, 1/2, k, α−1)-SuI;
4      foreach S ∈ S do
5          Q.append(S);
6          for i ← 1 to 2 log₂ n do
              /* Add a set of elements
                 from S that have 1 on
                 i-th least significant
                 bit of Balanced ID.    */
7              R_i(S) ← {v ∈ S : ⌊v/2^{i−1}⌋ = 1
                  mod 2};
8              Q.append(R_i(S));
9      ℓ ← ℓ/2;

10 T ← (n, 3δk/(α−1), k, α−1)-SSuI;
11 foreach T ∈ T do
12     Q.append(T);
13     for i ← 1 to 2 log₂ n do
14         R_i(T) ← {v ∈ T : ⌊v/2^{i−1}⌋ = 1
               mod 2};
15         Q.append(R_i(T));

16 return Q
```

---

$$\langle S_1^{(16)}, S_2^{(16)}, \ldots \rangle = \mathcal{S}^{(16)} = (64, 16, 1/2, 32, 16)\text{-SuI}$$

$$\langle S_1^{(32)}, S_2^{(32)}, \ldots \rangle = \mathcal{S}^{(32)} = (64, 32, 1/2, 32, 16)\text{-SuI} \qquad \langle T_1, T_2, \ldots \rangle = \mathcal{T} = (64, 8, 32, 16)\text{-SSuI}$$

$$\left\langle \underbrace{S_1^{(32)}, R\left(S_1^{(32)}\right), S_2^{(32)}, R\left(S_2^{(32)}\right), \ldots}_{\substack{\text{for decoding 16 (out of} \\ \text{32) hidden elements}}}, \underbrace{S_1^{(16)}, R\left(S_1^{(16)}\right), S_2^{(16)}, R\left(S_2^{(16)}\right), \ldots}_{\substack{\text{for decoding 8 (out of re-} \\ \text{maining 16) hidden elements}}}, \underbrace{T_1, R\left(T_1\right), T_2, R\left(T_2\right), \ldots}_{\substack{\text{for decoding the remaining 8} \\ \text{hidden elements}}} \right\rangle$$

Figure 1: An example of sequence of queries produced by Algorithm 1 for $n = 64, k = 32, \alpha = 17$. For simplicity we assume that $16 > \frac{3\delta k}{\alpha - 1} \geq 8$.

**Decoding algorithm.** During the decoding algorithm we process, in subsequent iterations, the feedbacks from enhanced selectors $\bar{\mathcal{S}}^{(\ell)}$ for $\ell = k, k/2, k/4, \ldots, 3\delta k/(\alpha - 1)$. We will later prove, by induction, that during processing $\bar{\mathcal{S}}^{(\ell)}$, $\ell/2$ new elements from $K$ are decoded. To show this, we consider any iteration and let set $K_1$ be the set of the elements that have been decoded in previous iterations while set $K_2 = K \setminus K_1$ be the set of unknown elements. We treat $K_1$ as the interfering set and, by the properties of $\mathcal{S}^{(\ell)}$, we know that for at least $\ell/2$ elements $v$, there exists a query $S \in \mathcal{S}^{(\ell)}$, such that $v \in S$, $|K_1 \cap S| < \alpha - 1$, $|K_2 \cap S| = 1$. We observe that since we already know the identifiers of all the elements from the interfering set $K_1$, then using feedbacks from the additional queries $\mathcal{R}(S)$ (corresponding to balanced IDs) we can exactly decode the identifier of $v$. We do this for all $\ell/2$ elements that are possible to decode in this iteration and we proceed to the next iteration. After considering all Selectors-under-Interference, we have only at most $3\delta k/(\alpha - 1)$ unknown elements. To complete the decoding we use a Strong-Selector-under-Interference, where the decoding procedure is exactly the same as in the case of SuI (the interfering set is also the set of already decoded elements). The properties of SSuI guarantee that we decode the identifiers of all the remaining elements from $K$. See the pseudocode of decoding Algorithm 2 for details of deocding for $\alpha > 1$. In the case where $\alpha = 1$ the decoding is straightforward, as the SSuI guarantees in this case no interference, hence the results (due to our enhancement $\bar{\mathcal{T}}$) contain exactly the identifiers of the elements from set $K$.

**Lemma 1.** *There is an explicit polynomial-time algorithm constructing non-adaptive queries $Q_1, \ldots, Q_m$ and decoding any hidden set $K$ of size at most $k \leq n$, from the feedback vector in polynomial time, under feedback $\mathcal{F}_\alpha$ and for $m = O((\frac{k}{\alpha})^2 \delta^2 \log^3 n + kd\delta \log^3 n)$ queries. Moreover, every element occurs in $O(\frac{k}{\alpha}\delta \log^2 n + d\delta \log^3 n)$ queries.*

*Proof.* We start from describing a procedure of revealing elements in any given set $K$ of at most $k$ elements, together with a formal (inductive) argument of its correctness. Our first goal is to show that by the beginning of $\bar{\mathcal{S}}^{(\ell)}$, for $\ell$ stepping down from $k$ to $3\delta k/(\alpha - 1)$, we have not learned about the identity of at most $\ell$ elements from the hidden set $K$.

The proof is by induction – it clearly holds in the beginning of the computation, as the set $K$ has at most $\ell = k$ elements. We prove the inductive step: by the end of $\bar{\mathcal{S}}^{(\ell)}$, at most $\ell/2$ elements are not learned. We set $K_2$ to be the set of learned elements and $K_1 = K \setminus K_2$. Clearly, $|K_2| \leq k$, and by the inductive assumption $|K_1| \leq \ell \leq k$. For such $K_1$ and $K_2$, by the definition of SuI, there are at most $\ell/2$ elements from $K_1$ that are not occurring in some round without other such elements or with at least $\alpha - 1$ of already learned elements from $K_2$. Consider a previously not learned element $v \in K_1$, for which there exists a *good query* in $\bar{\mathcal{S}}^{(\ell)}$, *i.e.,* a query $S \in \bar{\mathcal{S}}^{(\ell)}$ such that $S \cap K_1 = \{v\}$ and $|S \cap K_2| < \alpha - 1$. At this point of decoding of set $K$ we know the Balanced IDs of all elements from set $K_2$. Hence we can calculate the $2 \log_2 n$-bit feedback vector from sets $K_2 \cap R_1(S), K_2 \cap R_2(S), \ldots, K_2 \cap R_{2 \log_2 n}(S)$. We compare this feedback vector with the output of the enhanced selector, which is the feedback vector for sets $K \cap R_1(S), K \cap R_2(S), \ldots, K \cap R_{2 \log_2 n}(S)$. The difference between the latter and the former is exactly the Balanced ID of $v$. In case this difference does not form a Balanced ID of any element, *i.e.,* it has some value bigger than 1 or otherwise the number of 1's is different from $\log_2 n$, or in case $\mathcal{F}_\alpha(K \cap S) = \alpha$ (recall that $S$ is also in the constructed selector) the feedback from this $\mathcal{R}(S)$ is ignored. This is done to avoid misinterpreting the feedback and false discovery of an element which is not in $K$. Indeed, first note that the fact $|K \cap S| \geq \alpha$ will automatically discard the part of the feedback from $K \cap R_1(S), K \cap R_2(S), \ldots, K \cap R_{2 \log_2 n}(S)$, as it indicates that the intersection is too large to provide correct decoding of an element. Second, assuming $|K \cap S| < \alpha$, if there are no elements

---

**Algorithm 2:** Decoding of the elements for QGT with $\alpha$-capped feedback.

---

**Input:** Result($Q$) for queries $Q \in \mathcal{Q}$ constructed by Algorithm 1, based on SuI $\mathcal{S}^{(\ell)}$ for $\ell$ being powers of 2 stepping down from $k$ to $\frac{3\delta l}{\alpha-1}$ and SSuI $\mathcal{T}$.

1   $\ell \leftarrow k, K_{acc} \leftarrow \emptyset$ ;            /* Set $K_{acc}$ accumulates the decoded elements.   */

2   **while** $\ell > \frac{3\delta k}{\alpha-1}$ **do**    /* We decode $l/2$ elements in each iteration.*/

3      **for** $i \leftarrow 1$ **to** $l/2$ **do**

         /* Look for query for which we can decode a new element.      */

4         **foreach** $S \in \mathcal{S}^{(\ell)}$ **do**

5            **if** Result($S$) $< \alpha - 1$ **and** $|S \cap K_{acc}| = $ Result($S$) $- 1$ **then**

6              $K_{acc} \leftarrow K_{acc} \cup \{DecodeElement(S, K_{acc})\}$ ;

7      $\ell \leftarrow \ell/2$ ;

8   **while** $\ell > 0$ **do**    /* Iteratively decode all the remaining elements.*/

9      **foreach** $T \in \mathcal{T}$ **do**

10       **if** Result($T$) $< \alpha - 1$ **and** $|T \cap K_{acc}| = $ Result($T$) $- 1$ **then**

11         $K_{acc} \leftarrow K_{acc} \cup \{DecodeElement(T, K_{acc})\}$ ;

12         $\ell \leftarrow \ell - 1$ ;

13   **return** $K_{acc}$

---

1   **Procedure** *DecodeElement($Q, K_{acc}$)*

2      $v \leftarrow 0$ ;

3      **for** $j \leftarrow 2\log_2 n$ **downto** $0$ **do**

         /* Take the feedback from set $R_j(Q)$.   Calculate the feedback from
            set $R_j(Q)$, if hidden set was exactly $K_{acc}$.   The difference is
            $j$-th least significant bit of Balanced ID of new element $v$.     */

4         $v \leftarrow 2 \cdot v + $ Result($R_j(Q)$) $- |R_j(Q) \cap K_{acc}|$

5      **return** $v$

---

in $K_1 \cap S$ then the difference between feedbacks gives vector of zeros, and if there will be at least two elements in $K_1 \cap S$, the difference between feedbacks will contain a value of at least 2 or all 1's, as it will be a bitwise sum of at least two Balanced IDs of $\log_2 n$ ones each. By the definition of SuI we can find $l/2$ such elements $v$. This shows that during decoding of enhanced $\bar{\mathcal{S}}^{(\ell)}$ we learn the identities of $\ell/2$ new elements. This completes the inductive proof. Note here that the inductive step, being one of $O(\log n)$ steps, defines a polynomial time algorithm decoding some elements one-by-one – indeed, it computes two feedbacks of polynomial number of queries, computes the difference and deducts based on the structure of subsequent blocks of $O(\log n)$ size.

The above analysis implies, that before applying $(n, 3\delta k/(\alpha - 1), k, \alpha - 1)$-SSuI we have not discovered at most $3\delta k/(\alpha - 1)$ elements. Thus, by definition, $(n, 3\delta k/(\alpha - 1), k, \alpha - 1)$-SSuI combined with Balanced IDs reveals all the remaining elements in the same way as the SuI's above – the only difference in the argument is that instead of leaving at most $\ell/2$ undiscovered elements in the $\ell$-th inductive step, due to the nature of SuI's, the SSuI guarantees that *every* undiscovered element will occur in a good query. The same argument as for SuI's proves that the decoding algorithm defined this way works in polynomial time. By Theorem 3, the length of $(n, \ell, 1/2, k, \alpha)$-SuI is $O(\min \{n, \ell d\delta \log^2 n\})$, which sums up to $O(kd\delta \log^2 n)$, and is multiplied by $\Theta(\log n)$ due to amplification by Balanced IDs. By Theorem 5, the length of $(n, 3\delta k/\alpha, k, \alpha)$-SSuI is $O((\frac{k}{\alpha})^2 \delta^2 \log^2 n)$, and it is also increased by factor $\Theta(\log n)$ due to Balanced IDs. If we apply the above reasoning with respect to the number of queries containing an element, we get that every element occurs in $O(\frac{k}{\alpha}\delta \log^2 n + d\delta \log^3 n)$ queries.     $\square$

**Implementing $\alpha$-Round-Robin for large values of $k/\alpha$.** The question arises from the previous result if one could efficiently construct a shorter sequence of queries if $(k/\alpha)^2 > n/\alpha$? In the case of full feedback (*i.e.,* $\alpha = k$) the common way to deal with large values of $k$ is via Round-Robin, which means that queries are singletons and consequently, the length of such query sequence is $n$. This also works for an arbitrary value of $\alpha \leq k$, however the lower bound in Theorem 2 suggest that in such case there could exist a shorter query system of length $O((k + (n/\alpha)) \text{polylog } n)$. Indeed, if we modify our construction in such case, we could obtain such a goal. Namely, we concatenate:

- selectors $\bar{\mathcal{S}}^{(\ell)}$, for $\ell$ being decreasing powers of 2 from $\ell = k$ to $\ell = 6\delta k/(\alpha - 1)$;
- selectors $\bar{\mathcal{S}}|_{\alpha}^{(\ell)}$, for $\ell$ being decreasing powers of 2 from $\ell = 3\delta k/(\alpha - 1)$ to $\ell = 1$.

In the case, where $k > 6\delta k/(\alpha - 1)$, then the construction contains only selectors $\bar{\mathcal{S}}|_{\alpha}^{(\ell)}$. Then we enhance them based on Balanced IDs, as in the previous construction. Then, applying Theorem 3 for concatenated $\bar{\mathcal{S}}^{(\ell)}$ and Theorem 4 for concatenated $\bar{\mathcal{S}}|_{\alpha}^{(\ell)}$, instead of combination of Theorems 3 and 5 as it was in the proof of Lemma 1 with respect to $\bar{\mathcal{S}}^{(\ell)}$, we get the following result.

**Lemma 2.** *There is an explicit polynomial-time algorithm constructing non-adaptive queries $Q_1, \ldots, Q_m$ and decoding any hidden set $K$ of size at most $k \leq n$, from the feedback vector in polynomial time, under feedback $\mathcal{F}_\alpha$ and for $m = O((k + n/\alpha)d\delta \log^3 n)$ queries. Moreover, each element occurs in $O(d\delta \log^3 n)$ queries.*

*Proof.* The proof is analogous to the proof of Lemma 1, except that we continue proving the invariant until $\ell = 6\delta\kappa/(\alpha - 1)$, using same properties guaranteed by Theorems 3, and continue the invariant until $\ell = 1$, using SuI's of slightly different length formula from Theorem 4. The correctness argument, as well as polynomial-time query construction and decoding of the elements, are the same as in the invariant proof in Lemma 1. Then, by Theorem 3 for concatenated $\bar{\mathcal{S}}^{(\ell)}$ and by Theorem 4 for concatenated $\bar{\mathcal{S}}|_{\alpha}^{(\ell)}$, we argue that the total length of the obtained sequence is $m = O((k + n/\alpha)d\delta \log^3 n)$. Indeed, the first part results from the telescoping sum for different $\ell$ and the second component is a logarithmic amplification of the original $O((nd\delta/\alpha) \log n)$ length of SuI's; all is amplified by $O(\log n)$ due to enhancement of the used SuI's by Balanced IDs. In all the components, every element belongs to $O(d\delta \log n)$ queries, by Theorems 3 and 4, in the final sequence it occurs in $O(d\delta \log^3 n)$ queries. $\qquad\square$

Combining Lemma 1 with Lemma 2 gives Theorem 1. Note that in both lemmas the decoding algorithm proceeds query-by-query, each time spending polylogarithmic time on each of them; additionally, for each decoded element, an update of the feedback of next queries needs to be done, which takes time proportional to the number of occurrences of the discovered element in the queries. Thus, it is asymptotically upper bounded by the length of the sequence plus $k$ times the upper bound on the number of occurrences of an element in the queries (polylogarithmic).

## 5 Lower Bound

**Proof of Theorem 2.** We will first show the $\min\{n/\alpha, k^2/\alpha^2\}$ component. Assume that a sequence of queries $Q_1, Q_2, \ldots, Q_t$ of length $t$ solves Group Testing. We want to show the lower bound that holds for any feedback function capped at $\alpha$ hence we assume that the feedback function $\mathcal{F}$ returns the whole set (i.e., the identifiers of all the elements). Recall that $\mathcal{F}$ works only for sets with at most $\alpha$ elements. We begin by proving the following:

Claim A: For any set $K$, with $|K| \leq k$ and any $x \in K$, there must exist $\tau \in \{1, 2, \ldots, t\}$, such that $x \in Q_\tau$ and $|K \cap Q_\tau| \leq \alpha + 1$.

The proof is by contradiction. Assume that such a set $K^*$ and element $x^*$ exist for which there is no such query. Consider feedback vectors for sets $K^*$ and $K^* \setminus \{x^*\}$. For any query that does not contain $x^*$, the feedback is clearly identical. For any query $Q_\tau$, such that $x^* \in Q_\tau$, we have $|Q_\tau \cap K^*| \geq \alpha + 2$ and $|Q_\tau \cap (K^* \setminus \{x^*\})| \geq \alpha + 1$ and sets $K^*$ and $K^* \setminus \{x^*\}$ are indistinguishable under any feedback capped at $\alpha$ hence the sequence of queries does not solve the problem. This completes the proof of Claim A. $\blacksquare$

Take all queries that have at most $\alpha + 1$ elements and all elements that belong to such queries. We have: $N_s = \bigcup_{\tau \in \{1,2,\ldots,t\}|Q_\tau| \leq \alpha+1} Q_\tau$ and remaining elements $N_l = N \setminus N_s$. Consider two cases:

Case 1: $|N_s| \geq n/2$.

Observe that: $t \geq |\{\tau \in \{1, 2, \ldots, t\} : |Q_\tau| \leq \alpha + 1\}| \geq \frac{N_s}{\alpha+1} \geq \frac{n}{2(\alpha+1)}$.

Case 2: $|N_l| \geq n/2$.

In this case, we take an arbitrary subset $K_1$ of $k/2$ elements from $N_l$. For every element $x \in K_1$, we consider a set of queries $Q(x) = \{Q_\tau \in \{Q_1, Q_2, \ldots, Q_t\} : x \in Q_\tau, |Q_\tau \cap K_1| \leq \alpha + 1\}$. We show:

Claim B: For every $x \in K_1$, we have $|Q(x)| \geq \frac{k}{2(\alpha+2)}$. The proof is by contradiction. Assume that for some $x^* \in K_1$ we have $|Q(x^*)| < \frac{k}{2(\alpha+2)}$. Then, for every query $Q \in Q(x^*)$, we take

$\alpha + 2 - |Q \cap K_1|$ elements from $Q \setminus K_1$. Such elements exist since $|Q| \geq \alpha + 2$. Choose such elements for each query in $Q(x^*)$ and gather them in set $K_2$. Note that since $|Q(x^*)| < \frac{k}{2(\alpha+2)}$, then $|K_2| \leq k/2$. Now observe that set $K_1 \cup K_2$ and element $x^*$ violate Claim A. The obtained contradiction completes the proof of Claim B. ∎

Now observe that each query belongs to at most $\alpha + 1$ sets $Q(x)$ for different values of $x \in K_1$. Thus: $t \geq \frac{\sum_{x \in K_1} |Q(x)|}{\alpha+1} \geq \frac{k^2}{4(\alpha+1)(\alpha+2)}$. To complete the proof observe that any algorithm must fall either into Case 1 or Case 2, hence any algorithm needs to use $\Omega\left(\min\{n/\alpha, k^2/\alpha^2\}\right)$ queries.

To see that any algorithm in $\mathcal{F}_\alpha$ feedback model at least $k\frac{\log \frac{n}{k}}{\log \alpha}$ queries, observe that the feedback vector must be unique for each set $K$ with at most $k$ elements. Hence we need at least $\binom{n}{k}$ different feedback vectors for different sets. Feedback has at most $\alpha$ values hence we get $\alpha^t \geq \binom{n}{k}$ and $t \in \Omega(k\frac{\log \frac{n}{k}}{\log \alpha})$. This completes the proof of Theorem 2. ∎

## 6   Extensions, Applications and Open Directions

**Observations.**   The size of response to a capped quantitative query is not much bigger than beeping – $\log \alpha$ bits vs one bit of beeping – whereas the number of queries is smaller by a factor of $\alpha^2$ compared to beeping. Thus, even after normalization of the query complexity by the inverse of the size of the feedback, in the $\alpha$-capped model it is nearly $\alpha^2$ times better than the classic beeping, up to $\alpha = \sqrt{k}$.

Although optimizing the construction time is not our main focus, it is actually small. E.g., for $\alpha = \sqrt{k}$, the construction of all queries takes time $\widetilde{O}(n\sqrt{k})$. Most of the complexity comes from the construction of SSuI (Alg 4) – for each element of the universe, we consider $\widetilde{O}(\sqrt{k})$ zeros of some polynomial. We use a logarithmic number of SuIs in the construction, but each takes only $\widetilde{O}(n)$ time (Alg 3): in SuI for every element we list its neighborhood in disperser in polylogarithmic time [57] and take the Cartesian product with a polylogarithmic Reed-Solomon type of codeword.

**Fault-tolerance.**   Algorithms 1 and 2 are fault-tolerant with respect to $f \leq c \cdot \frac{k}{\alpha}$ *adversarially* jammed results, for some sufficiently small constant $c > 0$ – here by a jammed result we understand receiving a null as a result of a query. It also tolerates *stochastic* failures, where a result is jammed with probability $p \leq c$. Indeed, in adversarial failures we observe that, by dispersion, $c \cdot \frac{k}{\alpha}$ could be made negligible (by right selection of constant $c$) with respect to the number of selected elements in SuI, and it is also smaller than the number of single occurrences (i.e., without interference from other $k-1$ elements) of an element in SSuI. Thus the analysis of the algorithms hold (the original analysis have margins to accommodate those additional negligible jamming, as they were done with only asymptotic upper bounds). The tolerance of stochastic failures follows from the fact that a random $p$-fraction of results of queries coming from SuI's and SSuI are again negligible from perspective of the fraction of correctly encoded elements and can be accommodated by our asymptotic analysis.

**Theorem 6.** *Algorithms 1 and 2 tolerate* $f \leq c \cdot \frac{k}{\alpha}$ *adversarial jammings and stochastic jammings with probability* $p \leq c$, *for some sufficiently small constant* $c > 0$.

**Other applications.**   Applications to efficient GT and dynamic maintenance of multi-sets, graphs, as well as to Private Parallel Information Retrieval and coding, are detailed in Appendix B.

**Open directions.**   Considering only polynomially-constructible query systems leaves some interesting open directions. One such direction is whether optimal-length query sequence can be constructed in polynomial time or perhaps it is possible to show some reduction that constructing a nearly-shortest query sequence is computationally hard (even if we know that it exists). Shrinking polylogarithmic gaps between lower and (existential) upper bounds and improving constants is another challenging direction, as well as considering other interesting classes of GT models with an $\alpha$-capped results, e.g., parity. We also believe that with some adjustment, capped GT codes could be applied to efficiently solve many open problems in online streaming, communication and graph learning fields.

## Acknowledgments and Disclosure of Funding

Dominik Pająk was supported by the National Science Centre, Poland (NCN), grant UMO-2019/33/B/ST6/02988. Dariusz R. Kowalski was supported, in part, by the National Science Center, Poland (NCN), grant UMO-2017/25/B/ST6/02010.

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
