# OpenReview forum: "Scalable and Efficient Non-adaptive Deterministic Group Testing"
_NeurIPS.cc/2022/Conference — NeurIPS 2022 Accept_

### Official Review · Reviewer_4QpY · 2022-07-03

**Rating:** 3
**Confidence:** 3
**Soundness:** 2 fair
**Presentation:** 1 poor
**Contribution:** 1 poor

**Summary:**

In group testing, the goal is to identify $k$ defective objects among a total population of $n$ objects, by designing pools of objects for some pre-determined test, and then recovering the identities of the defective objects from the outcomes of the test. The standard notion of group testing reveals whether or not a defective object is included in a pool for each test. This paper considers quantitative group testing, where each test reveals the number of defective objects included in the test, up to a cap $\alpha$. The main result is a non-adaptive deterministic algorithm that uses roughly $\tilde{O}(k^2/\alpha^2)$ queries and decoding time. A nearly matching lower bound on the number of necessary queries is achieved for any non-adaptive algorithm.

**Questions:**

Are there specific applications for quantitative non-adaptive deterministic group testing algorithms? I think even the Private Parallel Information Retrieval (PPIR) application in Appendix B is a good motivation for quantitative group testing and in general non-adaptive group testing permits simultaneous queries instead of sequential queries, but I'm missing a situation where it would be good to have all of these demands simultaneously.

**Limitations:**

I do not see any potential negative societal impacts directly resulting from this work.

**Strengths And Weaknesses:**

The nearly matching upper and lower bounds are nice. Achieving an efficient decoding time is also an important aspect of the results in this paper. The techniques seem clearly above the bar for NeurIPS, but the current presentation is difficult for me to understand the proposed algorithm. For instance, it seems a key technical contribution is a combinatorial object called the selector under interference (SuI). Its construction is briefly described in appendix C using $(n,3\delta)$ strong selectors of Kautz and Singleton. However, neither the construction of the strong selector nor the relevant properties of the strong selectors are described. Thus the proof of correctness of the SuI is not accessible to me, e.g., statements such as line 700 invoke properties of the strong select that do not seemed to be discussed (please correct me if I am missing something!) Given that this is a purely theoretical paper, I think greater emphasis should be placed on the presentation, such as intuition for the individual components of the algorithm. For instance, the additional exposition in Section C.2 comparing the strong-selector-under-interference (SSuI) construction with a Reed-Solomon code was quite helpful.

More generally, I am concerned about the conceptual contribution of this paper within the context of NeurIPS. Although there is some precedence for group testing at NeurIPS, quantitative non-adaptive deterministic group testing seems significantly more limited. Indeed, I'm not sure that any of the motivations mentioned in this paper particularly require quantitative non-adaptive deterministic group testing, especially since randomized algorithms from previous work can achieve a smaller number of queries. Perhaps the authors can address this in their response.

---

### Official Review · Reviewer_gTi4 · 2022-07-11

**Rating:** 7
**Confidence:** 3
**Soundness:** 3 good
**Presentation:** 2 fair
**Contribution:** 3 good

**Summary:**

The goal in group testing is to identify a subset $K$ of $k$ elements from a universe $S$ of $n>k$ elements using queries. The queries considered in the paper are defined as the min between the size of the intersection of the query set $Q$ with $K$ and a parameter $\alpha$. The paper restricts the attention to deterministic non-adaptive algorithms. Prior work had considered the special cases $\alpha=1$ or $\alpha=k$ and gave algorithms with query complexity $k^2\log n$. The paper improves the dependence on $k$ by removing the square in the second case. It also works for the remaining range $1 \leq \alpha<k$ that had not been considered in prior work, matching the known upper bound for the case $\alpha=1$. The techniques for these upper bounds involve using strong selectors under interference (based on the prior work’s notion of disperses) and a strengthening of Round-Robin query systems. The paper complements the upper bounds with a lower bound, showing that the query complexity is optimal up to polylog factors. Finally, applications of main result regarding GT on multisets, online maintenance of dynamically changing graphs and private parallel information retrieval are briefly presented and discussed in Appendix.

**Questions:**

The points made in the paper regarding the completion of our understanding of deterministic non-adaptive GT from a theoretical viewpoint are clear. Without diminishing these contributions, there are some questions that come to mind:

* The upper bounds improve the dependence on $k$ in the query complexity but incur higher powers of $\log(n)$. Given that in practical scenarios we care about the number of queries made as an absolute number and even constant factors in query complexity matter (having to buy 5x COVID tests does matter) is there more empirical evidence to believe that the algorithm will outperform the previous ones in terms of number of queries made?

* Can you give some more comments on the benefits of having deterministic algorithms instead of randomized?


**Limitations:**

-

**Strengths And Weaknesses:**

Group testing is a problem that has been a topic of a long line of research. Its motivation by the concrete real-world applications relevant to the COVID-19 pandemic add to the significance of the problem. Therefore, the results of this paper are relevant to the theoretical CS community as well as being potentially useful in practical scenarios.

The writing is acceptable but with room for improvement. Overall, I found the style of writing to be dense and at parts hard to follow due to lack of intuition. Some more specific comments regarding Introduction:
* lines 52-56: At some points the presentation does not feel precise. The “gap” refers to the one between upper and lower bounds in the table? What exactly is the fault-tolerance interpretation of line 56?
* lines 67-74: It would be more useful to have a high-level sketch of the approach while explaining these concepts since just mentioning them does not provide much information to the new reader.
* Section 4 feels a bit dense.

I am not an expert in the topic/prior work. That being said, the arguments of the paper seem to require a certain degree of originality. To the extent that I checked the proofs I have not found issues on correctness.

---

> ### Author Response · Authors · 2022-08-02
> **response to the reviewer**
>
> REVIEWER: lines 52-56: At some points the presentation does not feel precise. The “gap” refers to the one between upper and lower bounds in the table? What exactly is the fault-tolerance interpretation of line 56?
>
> ANSWER: Yes, the “gap” refers to the multiplicative factor difference between upper and lower bounds for the best known results for explicitly constructed query systems. The mention of fault-tolerance refers to Section 6 where we initiate study and give preliminary results on fault-tolerant variant of the problem.
>
> REVIEWER: lines 67-74: It would be more useful to have a high-level sketch of the approach while explaining these concepts since just mentioning them does not provide much information to the new reader.
>
> REVIEWER: Section 4 feels a bit dense.
>
> ANSWER: Because of the page limit we could not fit all the details within the nine pages. But following the suggestion of the Reviewer we could shorten a bit the lower bound parts and move more details and intuitions on the algorithmic techniques from the Appendix to the main body.
>
> REVIEWER: The upper bounds improve the dependence on $k$ in the query complexity but incur higher powers of $\log(n)$. Given that in practical scenarios we care about the number of queries made as an absolute number and even constant factors in query complexity matter (having to buy 5x COVID tests does matter) is there more empirical evidence to believe that the algorithm will outperform the previous ones in terms of number of queries made?
>
> ANSWER: The reviewer is right, our methods so far yield larger constants and polylogarithms in the query complexity, mainly due to difficulties in explicit constructions of dispersers. However, any breakthroughs in constructions of dispersers will automatically lower the overhead to make our methods competitive even with randomized algorithms. Please also recall that we can not directly compare existing randomized solutions with our deterministic solution based only on the complexity, because our solution does not use any randomness and provides better guarantees -- works even against a strong adaptive adversary (creating an instance while observing how the algorithm builds the queries) and errors (see Theorem 6 in Section 6).
>
> REVIEWER: Can you give some more comments on the benefits of having deterministic algorithms instead of randomized?
>
> ANSWER: Randomized solutions could be indeed more tempting, but they require a large number of truly random bits (in practise, algorithms have access only to small truly random seeds, while randomized solutions typically have large number of $\Omega(n\log n \log k)$ random bits to construct the queries) and typically do not offer any provable guarantees in case when an adaptive adversary could create the set online during the learning process. Our deterministic solution has such guarantees and uses (asymptotically) very small number of queries if the feedback is reasonably rich. Hence, our solutions could be competing alternatives in systems using such feedback (e.g., communication systems with feedback based on more detail signal analysis) or against an ``active'' adversary.

---

### Official Review · Reviewer_NXCq · 2022-07-12

**Rating:** 6
**Confidence:** 3
**Soundness:** 3 good
**Presentation:** 3 good
**Contribution:** 3 good

**Summary:**

This paper considers a group testing where the goal is to identify all elements of an unknown set K of size k from large domain N of size n by asking a non-adaptive (capped) quantitative query where query result is the size of its intersection with the hidden set capped by some parameter \alpha. The main contribution of this paper is on proposing a polynomial-time construction algorithm constructing non-adaptive queries of length m=(\min(k^2/alpha,n/alpha)+k)*polylog n that allows polynomial-time decoding. By proving a lower bound on the length of queries, it is also shown that having the cap size alpha=\sqrt{k} is already optimal in achieving the shortest possible query complexity.

**Questions:**

* What are the possible applications/motivations of capped quantitative query? How effective such type of query is compared to simple binary beeping group testing when the sample complexity is properly normalized by the query complexity?

* Can the authors provide some intuitive explanation of why the cap size of \alpha=\sqrt{k} guarantees the optimality sample complexity (polylogrithmic number of queries per element) and why increasing \alpha does not provide much information gain that can change the order of sample complexity? Does this claim hold when one consider the normalized sample complexity (normalized by the query complexity)?

* Any suggestion to improve the poly(n) time complexity of query construction?


**Strengths And Weaknesses:**

Strengths:
* Proposes a polynomial-time construction algorithm constructing non-adaptive queries of length m=(\min(k^2/alpha,n/alpha)+k)*polylog n that allows polynomial-time decoding.

* Shows that having the capped query with cap size of alpha=\sqrt{k} allows the shortest-possible query complexity of size k\polylog n.

* Provides a lower bound demonstrating the optimality of the capped quantitative query with cap size of alpha=\sqrt{k}.

Weakness:

* Motivation of group testing with (capped) quantitative group query is somewhat missing. Group testing with beeping has been known to be prevalent in practical applications due to its simple binary output, but group testing with (capped) quantitative group query due is less motivated due to its complexity (since it requires more than a binary bit per query). Also, for a fair comparison of effectiveness of the capped quantitative group query, the sample complexity might need to be normalized by the number of information bits per query answer for the quantitative group query.

* The query construction complexity is poly(n) due to the complexity of disperser. Since the decoding requires only poly(k\log n) complexity, the main computational overhead comes from the query construction. Any suggestion to improve this time complexity?

* The query construction is based on techniques such as selectors-under-interference and strong-selectors-under-interference as well as dispersers and balanced IDs, of which the construction techniques are from [42], [44], [51]. It is unclear whether the construction directly follows from those previous results or there is some originality in the query construction part from this work.

---

> ### Author Response · Authors · 2022-08-02
> **response to reviewer**
>
> REVIEWER: Motivation of group testing with (capped) quantitative group query is somewhat missing.
>
> ANSWER: Motivations coming from more practical scenarios are given in Appendix B. Theoretical motivation is to better understand the transition between the two popular Group Testing models - beeping and QGT.
>
> R: Also, for a fair comparison of effectiveness of the capped quantitative group query, the sample complexity might need to be normalized by the number of information bits per query answer for the quantitative group query.
>
> A: We discuss this interesting point later.
>
> R: The query construction complexity is poly(n) due to the complexity of disperser. Since the decoding requires only $\mbox{poly}(k\log n)$ complexity, the main computational overhead comes from the query construction. Any suggestion to improve this time complexity?
>
> A: Our lower bound and the logarithmic lower bound on left-degree of any disperser suggest that the overhead improvement is possible only up to $\log n$.
>
> R: The query construction is based on techniques such as selectors-under-interference and strong-selectors-under-interference as well as dispersers and balanced IDs, of which the construction techniques are from [42],[44],[51]. It is unclear whether the construction directly follows from those previous results or there is some originality in the query construction part from this work.
>
> A: Our novelty regarding selectors is twofold:
> - We identified additional properties of selectors (e.g., avoiding interfering set with capped intersections), that could not be formally deducted from the previous definitions, but allow to improve the solution to QGT
> - We applied careful mathematical analysis to choose specific parameters of dispersers and codes and to prove that new properties hold
>
> R: What are the possible applications/motivations of capped quantitative query? How effective such type of query is compared to simple binary beeping group testing when the sample complexity is properly normalized by the query complexity?
>
> A: Examples are: wireless communication (decoding the number of at most $\alpha$ interfering devices from the signal); bio-chemical testing with capped scale; streaming algorithms with limited processing time.
>
> Observe that the answer to a quantitative query is not much more complex than beeping since our model uses $\log\alpha$ bits vs one bit of beeping, whereas the number of queries decreases by a factor of $\alpha^2$ compared to beeping. Thus, even after normalization, the query complexity in the $\alpha$-capped model is nearly $\alpha^2$ times better than the classic beeping, up to $\alpha=\sqrt{k}$.
>
> R: Can the authors provide some intuitive explanation of why the cap size of $\alpha=\sqrt{k}$ guarantees the optimality sample complexity (polylogrithmic number of queries per element) and why increasing $\alpha$ does not provide much information gain that can change the order of sample complexity? Does this claim hold when one consider the normalized sample complexity?
>
> A: $\alpha=\sqrt{k}$ seems to be a sweet spot balancing two different phenomena -- superimposed coding and dispersion.
> Enriching the beeping model ($\alpha$ increases starting from $1$) allows to improve the superimposed coding (used for the beeping feedback), and the improvement is roughly quadratic in terms of $\alpha$, due to the nature of superimposed codes. On the other hand, limiting the full quantitative
> model ($\alpha$ decreases starting from $k$), which admits polylogarithmic number of queries, increases the number of queries roughly linearly in $\alpha$, as the nature of dispersion is linear. Both these phenomena equalize query complexities at $\alpha=\sqrt{k}$. This is illustrated in our construction -- in the beginning, we are making progress relying on dispersion-based SuI's for small $\alpha$, then with the growth of $\alpha$ the length of used SuIs becomes similar to the length of the code-based SSuI; it happens for $\alpha=\sqrt{k}$. The SSuI allows to discover all remaining elements and  finishes the learning process. The last fact explains why further increase of $\alpha$ is not needed. Regarding normalized sample complexity, based on our answer to the previous question, we claim that roughly the same phenomenon holds.
>
> R: Any suggestion to improve the poly(n) time complexity of query construction?
>
> A: Although optimizing the construction time is not our main focus, it is actually quite small. The construction of all queries takes time $O(n\sqrt{k}\mbox{ polylog }n)$. Most of the complexity comes from the construction of SSuI (Alg 4) -- for each element of the universe, we consider $O\sqrt{k}\mbox{ polylog } n)$ zeros of some polynomial. We use a logarithmic number of SuIs in the construction, but each takes only $O(n\mbox{ polylog }n)$ time (Alg 3): in SuI for every element we list its neighborhood in disperser in polylogarithmic time [51] and take the Cartesian product with a polylogarithmic Reed-Solomon type of codeword.

---

### Official Review · Reviewer_KFv3 · 2022-07-12

**Rating:** 5
**Confidence:** 3
**Soundness:** 2 fair
**Presentation:** 2 fair
**Contribution:** 2 fair

**Summary:**

In this work, the authors look at the problem of group testing where we are interested to learn about a hidden subset K of size k of a universe N of size n >> k. The authors assume access to subset queries that return its intersection size with K or $\alpha$ if the latter exceeds $\alpha$. The special cases $\alpha=1$ and $k$ have been extensively studied before, while the general case has not been studied so far. The authors give a non-adaptive deterministic algorithm with an almost optimal query complexity.

caveat: I did not go through the proofs or technical lemmas in great detail.

**Questions:**

None.

**Limitations:**

None.

**Strengths And Weaknesses:**

Strength: The problem is interesting and the presented results are significant.

- previous works only looked at the special cases $\alpha=1$ and $k$, whereas they consider the general problem
- their upper and lower bound theorems are significant as they match up to small factors
- interestingly they observe that the query complexity does not change by much as $\alpha$ grows from $\sqrt{k}$ through $k$ (LN# 134)
- they give new results for certain combinatorial objects (SuI, SSI, dispersers)
- their algorithms do not use any randomization, adaptiveness, or unlimited computational power, which is harder for upper bound

Weakness:

- the results are not supported by experiments (soundness).
- a tighter query complexity characterization, even though not in polytime, is lacking (LN# 379) (contribution)
- small examples to motivate/explain their algorithms or insights would have made the paper more readable (presentation)

---

### Meta-Review · Area_Chair_PNp3 · 2022-08-20

**Recommendation:** Accept
**Confidence:** Less certain

**Metareview:**

This is quite a borderline case.  Three of the reviewers are in favor of acceptance with three different degrees (borderline/weak/clear), while one reviewer maintains (clear) rejection.

However, the content of that review and the remarks during the discussion period are perhaps better aligned with a score of 4 (borderline reject) than 3.  For example, the reviewer explicitly states that the paper is above the bar in terms of technical content.

The main concerns were then (i) suitability for NeurIPS, (ii) scope of contributions, and (iii) clarity of presentation, including technical content.

Regarding (i), although there exist conferences that are a better match to the paper, the topic does seem to be an OK match for NeurIPS, which often contains papers in high-dimensional statistical topics, e.g., compressive sensing.  (And *some* group testing papers have appeared before, albeit very rarely.)

Regarding (ii), the combination of "deterministic + non-adaptive + quantitative" is a fairly specific one, but "deterministic + non-adaptive" is a very widely sought-after combination in the group testing literature, so I believe the overall combination isn't farfetched by any means.  It would be ideal to have a clear and specific practical setting with this motivation, but being a theory paper, this doesn't seem like a deal-breaker.  Moreover, the general consensus (albeit with varying degrees of enthusiasm) is that the results derived are valuable.

Perhaps the main aspect that makes the decision borderline is (iii).  The reviewer asked some specific questions about the SuI definition and intuition, and the authors' response did not address those questions.  The paper is very technical and would potentially be much easier to follow if it were presented in a more accessible manner, e.g., with more aid via diagrams, discussions, intuition, additional cross-references & hints, and so on.   I agreed with this sentiment after stepping through one of the proofs -- I found it to be readable but with quite a bit of difficulty (e.g., pausing for a long time to check a step that was made out to be very simple).  This could certainly be detrimental to how much non-experts can gain from reading the paper.

On the more positive side, based on the proof that I read and similar efforts by another reviewer, the analysis appears to be correct, at least as far as we can tell.  Overall, I will cautiously judge that the good technical contributions and apparent correctness should outweigh the room for improvement in accessibility.  In any case, the authors should very carefully revise the paper with the reviews and the above comments in mind.



**Award:**

No

---

### Decision · Program_Chairs · 2022-09-14

Accept